# Cooperative Optimization of Electric Vehicles and Renewable Energy Resources in a Regional Multi-Microgrid System

**Jin Chen**, **Changsong Chen** *** and **Shanxu Duan**

State Key Laboratory of Advanced Electromagnetic Engineering and Technology, School of Electrical and Electronic Engineering, Huazhong University of Science and Technology, Wuhan 430074, China; eufoniuso@163.com (J.C.); duanshanxu@hust.edu.cn (S.D.)
* Correspondence: ccsfm@163.com



**Featured Application: The main goal of this work is to provide a cooperative optimization method for regional multi-microgrid system, optimizing dispatching strategy and capacity allocation of electric vehicles and renewable energy sources. Meanwhile, across-time-and-space energy transmission of electric vehicles is considered in the optimization model.**

**Abstract:** By integrating renewable energy sources (RESs) with electric vehicles (EVs) in microgrids, we are able to reduce carbon emissions as well as alleviate the dependence on fossil fuels. In order to improve the economy of an integrated system and fully exploit the potentiality of EVs' mobile energy storage while achieving a reasonable configuration of RESs, a cooperative optimization method is proposed to cooperatively optimize the economic dispatching and capacity allocation of both RESs and EVs in the context of a regional multi-microgrid system. An across-time-and-space energy transmission (ATSET) of the EVs was considered, and the impact of ATSET of EVs on economic dispatching and capacity allocation of multi-microgrid system was analyzed. In order to overcome the difficulty of finding the global optimum of the non-smooth total cost function, an improved particle swarm optimization (IPSO) algorithm was used to solve the cooperative optimization problem. Case studies were performed, and the simulation results show that the proposed cooperative optimization method can significantly decrease the total cost of a multi-microgrid system.

**Keywords:** electric vehicles; renewable energy sources; microgrid; economic dispatching; capacity allocation; cooperative optimization

## 1. Introduction

Due to the characteristics of low carbon emission and sustainability, renewable energy sources (RESs) have attracted much attention in recent years [1–3]. In order to cope with the intermittency and uncertainty of RESs, the concept of microgrid (MG) was proposed to increase the utility of RESs by integrating them with energy storage units [4]. However, available storage units are expensive and a massive usage of them would significantly add to the cost of operation. The complexity of a microgrid's controlling is also an issue which challenges the traditional ways a power system operates.

However, increasing of electric vehicles' (EV) penetration brings more uncertainty to the operation of a power system. Uncontrolled charging behaviors of EV owners in load peak hours aggravate the burden of a power grid, posing great threats to the grid's operation.

Since EVs have the ability of storing energy, they can serve as storage units in RES-equipped power systems. Taking EVs as energy storage units and integrating them with RESs can attenuate

the burden of disorderly charging while reducing the installation of expensive storage batteries [5]. Many studies focusing on the integration of EVs and RESs have been published in recent years; the main topics of these can be briefly separated into two categories, economic dispatching and system capacity allocation.

Most research has discussed the issue from the aspect of economic dispatching. The majority of the studies elaborated the optimal charging management of EVs [6,7]. However, as a kind of storage units, EVs can also discharge to the main grid or microgrid, providing a vehicle-to-grid (V2G) service to participate into the operation of the power systems [8,9]. Dispatching EVs which provide V2G services was also discussed in many papers. In [10], a MILP (Mixed-integer linear programming) model is proposed to find the optimal charging and discharging strategy of hybrid electric vehicles (HEVs). The proposed method in the paper improved the automation of HEV navigation systems and find a way to a more economic and sustainable transportation system. In [11], an integrated microgrid system considering both renewable energy source and electric vehicles is constructed. EV owners are regarded as a kind of flexible load of the demand response (DR), and the proposed optimization method significantly reduced the cost of the system. Though there is a large number of works that focus on the V2G service of EVs, some of them achieving significant progress, there are still some issues to be addressed. Most models proposed in the works are applied on only single microgrids. Interaction among different microgrids is not considered. On the other hand, the mobility of EVs is not fully exploited, and the across-time-and-space energy transmission of EVs in not considered.

Capacity allocation of the integration system is also an important issue in the optimization of the system, but not frequently mentioned in research. Existing works focus only on the planning of BESSs (Battery Energy Storage Systems) and RESs. Most of the research allocated either charging devices or RESs rather than cooperatively optimizing them at the same time. In [12], planning different types of plug-in electric vehicles (PEV) that charge infrastructures were studied. On the other hand, in [13], capacities of RESs were optimized to minimize the whole system's investment cost while meeting the load demand. In [14], an algorithm for microgrid planning was proposed considering massive connection of EV charging demands, and the system investment cost as well as $CO_2$ emission is reduced. However, on the one hand, the relationship between economic dispatching and capacity configuration of the microgrid system is not considered in the papers. On the other hand, only a few of the studies discuss the allocation for both RESs and EVs, and the concept of cooperative optimization is not found in any research.

To sum up, there has already been much research on the topic of optimal dispatching and system allocation of RES-EV integrated systems [15,16]. Nevertheless, EV's ability of across-time-and-space energy transmission (ATSET) in the context of multi-microgrid systems is not discussed in the literature. The relationship between system economic dispatching and capacity configuration is not considered, and cooperative optimization method considering both allocation and dispatching are not mentioned in the studies. On the one hand, an allocation without system dispatching would cause a redundant installation of RESs and increase the total cost of the system. On the other hand, EV dispatching without optimal allocation of the system would decrease the efficiency of their integration into the microgrid system. Therefore, a cooperative optimization that simultaneously considers allocation and dispatching is necessary for the system's economic operation. For the reference, an optimization method using both the configuration of RESs and EVs is not considered in most of the works.

In order to fully exploit the potential of EV's mobile storage ability as well as reduce the redundant installation of RESs, a cooperative optimization considering EV's across-time-and-space energy transmission is presented in this paper. Both the installation capacity of RESs and the number of EV charging/discharging infrastructures (EVCDIs) are considered as allocation optimization variables.

The main contributions of this paper are listed as follow:

1.  An integration mechanism of RESs and EVs is illustrated, and a mathematical model of the V2G-integrated regional multi-microgrid system is established;

2.　The concept of EV's ATSET is illustrated. Based on the ability of EV's ATSET, a cooperative optimization method for economic dispatching and system capacity allocation is proposed;

3.　To fully exploiting the potential of the EV's mobile energy storage ability, both the installation capacity of RESs and the number of EVCDIs are optimized in the cooperative optimization process.

The following manuscript is organized as follows. In Section 2, basic concept of cooperative optimization is illustrated. Mathematical model of a typical regional multi-microgrid system is constructed and elaborated in Section 3. In Section 4, objective functions and restrictions are described. The cooperative optimization model is demonstrated in Section 5 and case studies are performed in Section 6. Finally, in Section 7, conclusions are drawn.

## 2. Concept of Cooperative Optimization

The theoretical structure of the cooperative optimization method proposed in this paper can be illustrated as in Figure 1.

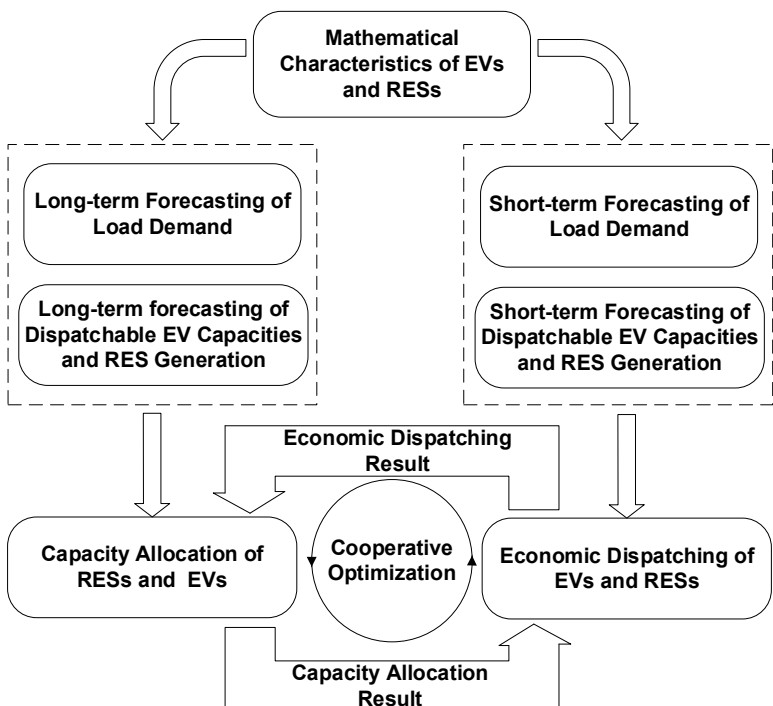

**Figure 1.** Brief process of cooperative optimization of EVs (Electric Vehicles) and RESs (Renewable Energy Sources).

The model can be separated into two sections: one is the forecasting of load demands and RES generations—the accuracy of the forecast can significantly influence the availability of the optimization results—and the other is the main part of the cooperative optimization. In the optimization process, forecasting data are derived according to the mathematical character of EVs and RESs, sent to the dispatching and allocation sections. The dispatching model will dispatch the operation of microgrid system including EVs and RESs, and the allocation section can optimize the installation capacities of EVBCDIs (Electric Vehicle bidirectional charging/discharging infrastructure) and RESs. An allocation module generates the initial system allocation. A dispatching model then optimizes the operation of the system according to the system allocation and feeds the results back to the allocation module. The system allocation is then updated by an allocation module according to the data from the dispatching module. Through several repetitions of this interaction, both allocations and dispatching of the system are finally optimized. Details of the cooperative optimization process will be discussed in following sections.

In this paper, three respects of cooperative optimization are considered.

Firstly, RESs and EVs are cooperatively optimized. By optimally dispatching EVs, energy generated by RESs can be redistributed across different times and spaces, and the utilization efficiency of RESs can be improved. Furthermore, EVs can also profit from interacting with RESs and participating into the operation of the system.

Secondly, microgrids in the regional multi-microgrid system are cooperatively optimized. With the ATSET of EVs, energy can be transmitted through different microgrids, and microgrids with lower electricity prices indirectly sell their energy to microgrids with higher electricity prices. Through such a cooperation, microgrids with lower prices can sell energy to make profits, and microgrids with higher prices can purchase electricity form EVs with lower prices.

Thirdly, economic dispatching and system allocation are cooperatively optimized. An economic dispatching of the system can reduce the cost of a redundant installation of RESs and EVCDIs, and an appropriate system allocation can in turn better exploit the potential of generation and energy storage units to serve the load with less costs.

## 3. Mathematical Model

### 3.1. Structure of the Multi-Microgrid System

The structure of a classic multi-microgrid system is demonstrated as follows in Figure 2. The system is constructed with several independent microgrids, which can be briefly categorized into residential microgrids (RMGs) and office building microgrids (OBMGs). The load curves of these two kinds of microgrids are slightly different, and OBMGs have a heavier load demand in general.

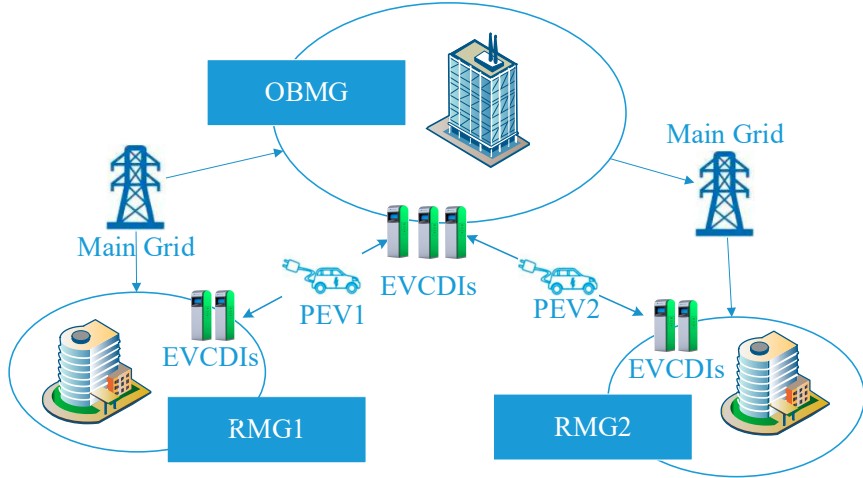

**Figure 2.** Structure of a classic multi-microgrid system.

In every single microgrid, photovoltaic (PV) generation modules and domestic small-size wind turbines (WTs) are installed according to the microgrid's load demand. To satisfy the requests for charging of both residents and staff, charging plots were installed in all microgrids, and parts of them include EVBCDIs for participating in the power system's operation. For the sake of power balance and the stability of system's operation, all microgrids are connected to the main grid through connection transformers.

### 3.2. Renewable Energy Sources

Most load demand of microgrid systems are supplied by distributed energy sources, the majority of which are the RESs mentioned in this paper. Since RESs' characters are highly relevant to the weather and environment, their generations are volatile and intermittent. For the sake of safety and availability, an accurate and reliable prediction of RES generation is necessary. In this paper, artificial

neural networks (ANNs) are adopted as the prediction model, and two models from [17,18] are used in the proposed optimization model. Due to the restriction of the paper length, details of the two models are not demonstrated here.

### 3.3. Electric Vehicles

The multi-microgrid system considered in this paper is a living–working combined system. Residents are all staff from office building areas. Thus, all the EVs considered in this paper are commuting cars. They park in the RMG to charge during the knock-off hours from 19:00 to 8:00 and park in the OBMG during working time from 9:00 to 18:00. Considering the random behaviors of the staff, the arrival and departing times of EVs were derived by Monte Carlo experiment and random variables were then added.

Energy variation of EV batteries came from two aspects. One is the energy consumption of EVs for their driving on the commuting route, and the other is charging and discharging through EVBCDIs. State of capacity (SOC) was used to present the energy remaining in the mobile batteries. During the driving distance, the SOC of PEVs could be calculated as (1).

$$SOC_{i,j}^{EV}(t) = SOC_{i,j}^{EV}(t - d_{i,j}) - D_{i,j}^{EV} \cdot C_d \tag{1}$$

where $SOC_{i,j}^{EV}(t)$ is the SOC of the $j$th EV in the $i$th microgrid in the $t$th hour, and $d_{i,j}$ is the driving time. $C_d$ is the energy consumption rate of EV's driving, and $D_{i,j}^{EV}$ is the driving distance. Since EV's driving distances are slightly different due to each owner's random behaviors, the distance was also generated from Monte Carlo experiment, and can be demonstrated as Equation (2) [19].

$$D_{i,j}^{EV} = E(D_{i,j}^{EV}) + k_{i,j} \cdot \sigma_{i,j}^{EV} \tag{2}$$

where $D_{i,j}^{EV}$ is the actual value of $D_{i,j}^{EV}$; $E(D_{i,j}^{EV})$ is the mathematical expectation of the driving distance; $k_{i,j}$ is a random parameter and is normally distributed; $\sigma_{i,j}^{EV}$ is the standard deviation of the driving distance. Notice that according to the statistical data of automobile's driving, the driving distance is also normally distributed.

When parking in the OBMG or RMG, the SOCs of EVs were determined only by the charging and discharging behaviors. Considering the self-discharging of EV batteries, the SOC can be calculated according to Equation (3).

$$\begin{cases} SOC_{i,j}^{EV}(t) = SOC_{i,j}^{EV}(t-1) \cdot (1 - \delta_{SD}) \\ \qquad - P_{i,j}^{EV}(t) \cdot \Delta t \cdot \eta_C & if\ P_{i,j}^{EV}(t) < 0 \\ SOC_{i,j}^{EV}(t) = SOC_{i,j}^{EV}(t-1) \cdot (1 - \delta_{SD}) \\ \qquad - P_{i,j}^{EV}(t) \cdot \Delta t / \eta_D & if\ P_{i,j}^{EV}(t) > 0 \end{cases} \tag{3}$$

where $P_{i,j}^{EV}(t)$ is the charging and discharging power of the $j$th EV in the $i$th microgrid in the $t$th hour, while $P_{i,j}^{EV}(t) < 0$ for charging and $P_{i,j}^{EV}(t) > 0$ for discharging. $\delta_{SD}$ is the self-discharging rate. $\eta_C$ and $\eta_D$ are the charging and discharging efficiency rates, respectively.

### 3.4. Operation Mechanism

As illustrated in Section 3.1, the system is mainly formulated by EVs, RESs, EVBCDIs, and other assistant devices as connection transformers. In order to optimize the system's operation, data of SOC are collected by the intelligent meters installed in the EVBCDIs and then uploaded to the microgrid management system (MMS) and finally to the general management system (GMS) of the whole multi-microgrid system. The GMS optimizes system operation data and then send the dispatching

results to the EVBCDIs to be performed. A brief description of the operation mechanism for the cooperative optimization is demonstrated as Figure 3.

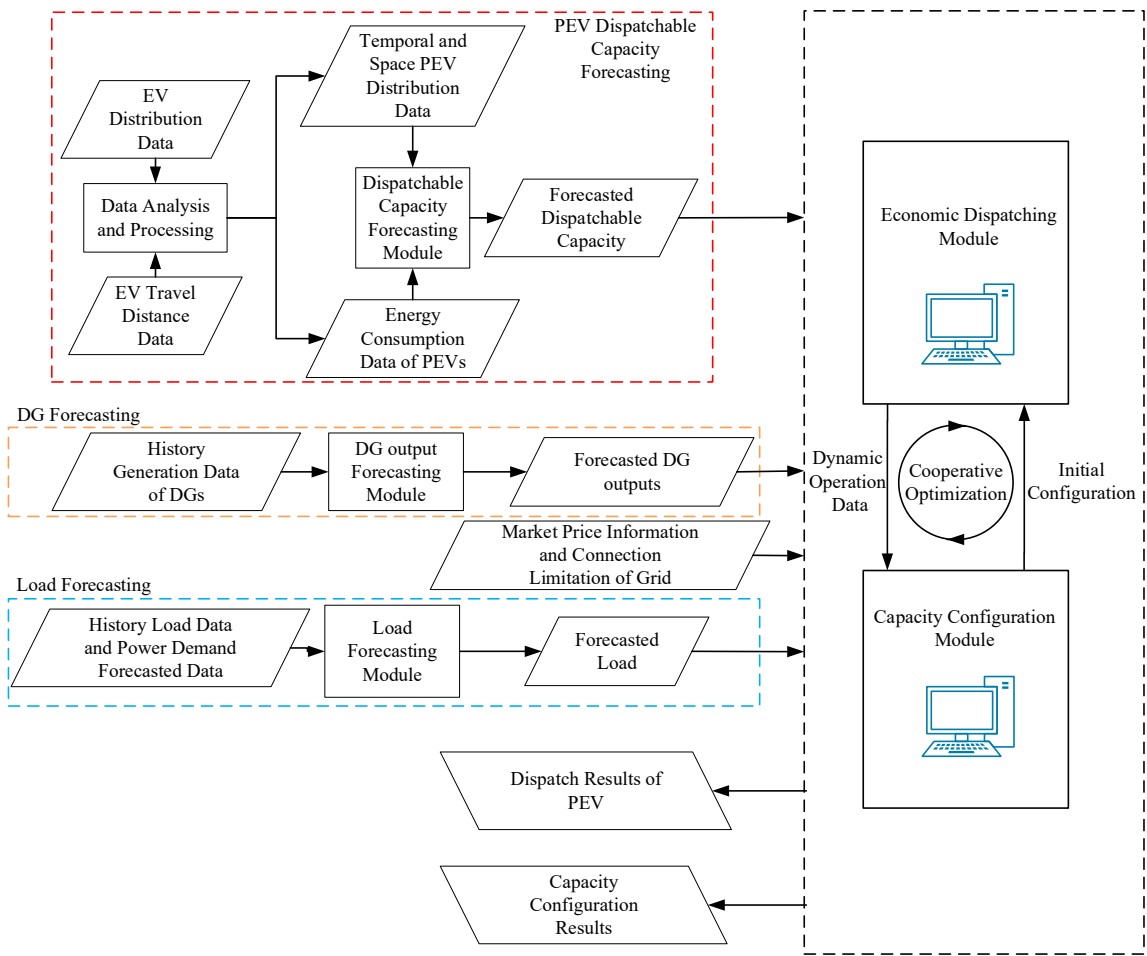

**Figure 3.** Mechanism of the cooperative optimization.

Since there are many EVs in each microgrid, it is not impractical to optimize the charging and discharging powers of every EV, nor is it necessary. In this paper, a hierarchical structure of the management system is utilized. EVs are separated into several EV fleets according to the residential areas they belong to, and each fleet is consisted of all the EVs form the same RMG. Charging and discharging powers of EV fleets rather than single EVs are optimized by GMS, and the data are sent to the management systems of EV fleets. The power of single EVs is then optimized according to the data from the GMS. In this paper, only the optimal dispatching of EV fleets is discussed, and the dispatching method of single EVs can refer to [20] and is not comprehensively discussed in this work.

## 4. Optimization Problem Formulation

As a typical mathematical programming problem, objective function and optimization constraints are significant components in the problem construction. The objective function and constraints of the cooperative optimization studied in this paper are formulated as follows.

### 4.1. Objective Function

The main optimization object is to improve the economics of the system allocation and decrease the cost of the microgrid operator. The multi-microgrid system discussed in the proposed model, which consists of several microgrids, is assumed to be owned by one single operator. Therefore, minimizing

the total operation cost of the whole system is regarded as the objective function. The total cost can be described by Equations (4)–(7).

$$C_{MMS} = C_{RMG} + C_{OBMG} \tag{4}$$

$$C_{RMG} = \sum_{r=1}^{N_{RMG}} C_r^{RMG} \tag{5}$$

$$C_{OBMG} = \sum_{o=1}^{N_{OBMG}} C_o^{OBMG} \tag{6}$$

$$N_{MG} = N_{RMG} + N_{OBMG} \tag{7}$$

where $C_{MMS}$ is total cost of the whole multi-microgrid system, $C_{RMG}$ and $C_{OBMG}$ are the total costs of RMGs and OBMGs, and $C_r^{RMG}$ and $C_o^{OBMG}$ are the costs of the *r*th RMG and the *o*th OBMG. $N_{MG}$ is the number of all microgrids in the system. $N_{RMG}$ and $N_{OBMG}$ are numbers of RMGs and OBMGs.

The cost of each microgrid includes the costs of wind generation, solar generation, EVs and the cost of exchanging power with the main grid. It is calculated by Equation (8).

$$C_i^{MG} = C_i^{PV} + C_i^{WT} + C_i^{EV} + C_i^{G} \tag{8}$$

where $C_i^{PV}$, $C_i^{WT}$, $C_i^{EV}$ and $C_i^{EV}$ are costs of PV generation, wind power, EV operation, and energy exchanging with the main grid. Costs of PV generation and wind power are formulated by capital costs and costs of maintenance—they are constant values. Therefore, the costs of PV and wind generation can be calculated as Equation (9).

$$\begin{cases} C_i^{PV} = C_{PV} \cdot N_i^{PV} \\ C_i^{WT} = C_{WT} \cdot N_i^{WT} \end{cases} \tag{9}$$

where $C_{PV}$ and $C_{WT}$ are costs of single PV module and wind turbine, and $N_i^{PV}$ and $N_i^{WT}$ are numbers of PV modules and wind turbines in the *i*th microgrid.

Since the battery will depreciate with the increasing of charging/discharging cycles, extra charging and discharging caused by V2G services can significantly raise the cost of EV owners. In order to compensate for the EV owners, the extra cost caused by V2G should be considered. Considering depreciation expense, the cost the EVs can be calculated by Equation (10).

$$C_i^{EV} = N_i^{EV} \cdot C_{EVBCDI} + \sum_{t=1}^{T} \sum_{j=1}^{N_i^{EV}} P_{i,j}^{EV}(t) \cdot p^{EV}(t) + \sum_{j=1}^{N_i^{EV}} C_{depi,j}^{EV} \tag{10}$$

where $N_i^{EV}$ is the number of EVs in the *i*th microgrid, $T$ is the optimization period, $C_{EVBCDI}$ is the cost of EVBCDI, $P_t^{EV}$ is the price of EV charging and discharging, and $C_{dep}^{EV}$ is the depreciation cost which is related to the charging/discharging cycles and is calculated according to [21].

Most of the energy in the main grid is generated from thermal plants, which produce large amount of pollutant as $CO_2$ and $SO_2$. In order to produce as little pollution as possible and to increase the utilization rate of RESs, energy exchanging with the main grid should be restricted. Therefore, in this paper, an environmental penalty is introduced. The costs of exchanging energy with the main grid can be calculated by Equation (11).

$$C_i^{G} = \sum_{t=1}^{T} p^G(t) \cdot P_i^G(t) + \sum_{t=1}^{T} C_i^{Gpen}(t) \tag{11}$$

where $p^G(t)$ is the prices of trading energy with the main grid, $P_i^G(t)$ is the exchanging power of the $i$th microgrid with the main grid in the $t$th hour, and $C_i^{Gpen}(t)$ is the environmental penalty of the $t$th hour.

$$C_i^{Gpen}(t) = [\max(0, P_i^G(t))] \cdot k_{env} \tag{12}$$

where $k_{env}$ is the pollution factor, and it is determined by the types and proportion of pollutants [22].

*4.2. Constraints*

Some restrictions exist in the real system due to the characteristic of power devices and system balance. The constraints of the optimization problem in this paper are listed as follows.

4.2.1. Balance of Power Supply and Demand

The power generated and supplied by the microgrid should meet the power demand of end users in every dispatching time interval.

$$P_i^{EV}(t) + P_i^{WT}(t) + P_i^{PV}(t) + P_i^G(t) = P_i^L(t) \tag{13}$$

where $P_i^{PV}(t)$, $P_i^{WT}(t)$, and $P_i^L(t)$ are power of PV generation, wind generation, and load demand respectively.

4.2.2. Capacity Constraint of EV Batteries

Overcharge and discharge influence the life span of the batteries. However, in order to guarantee the driving demand of EV owners, SOC of EVs should be always higher than a certain minimum value.

$$SOC_{\min}^{EV} \le SOC_{i,j}^{EV}(t) \le SOC_{\max}^{EV} \tag{14}$$

4.2.3. Power Limits

Due to the physical limitations of EVBCDIs and connection transformers, both charging/discharging power and exchanging power are limited to a certain range.

$$\begin{cases} P_{\min}^{EV} \le P_{i,j}^{EV} \le P_{\max}^{EV} \\ P_{\min}^G \le P_{i,j}^G \le P_{\max}^G \end{cases} \tag{15}$$

4.2.4. Installation Constraints

To satisfy the charging demand of all EV users in the microgrid system, a total number of EVBCDIs in RMGs and the number of that in OBMGs should be equal, as demonstrated in (16) and (17).

$$\sum_{i=1}^{N_{MG}} N_i^{EV} = \sum_{r=1}^{N_{RMG}} N_r^{EVR} = \sum_{o=1}^{N_{OBMG}} N_o^{EVO} \tag{16}$$

$$N_{MG} = N_{RMG} + N_{OBMG} \tag{17}$$

where $N_{RMG}$ and $N_{OBMG}$ are numbers of RMGs and OBMGs, and $N_r^{EVR}$ and $N_o^{EVO}$ are the numbers of EVBCDIs in the $r$th RMG and the $o$th OBMG.

## 5. Two-Loop Optimization

*5.1. Improved Particle Swarm Optimization*

Since the optimization problem is a non-linear problem with a complex formulation of objective function, classic mathematical programming is not a proper method to solve it. Instead, an improved

particle swarm optimization algorithm (IPSO) is used in this paper. In IPSO, each potential solution is regarded as a particle, and during the optimization, the particles update their positions and velocities in each loop iteration. After several rounds of iteration, the global best value of all the particles is adopted as the optimal solution.

As a kind of intelligent algorithm, IPSO can efficiently search for the optimal solution of a complex nonlinear optimization problem. Details of the algorithm can be seen in [23,24].

### 5.2. Two-Loop Optimization Process

The cooperative optimization has a two-loop structure. The process is demonstrated as follows in Figure 4.

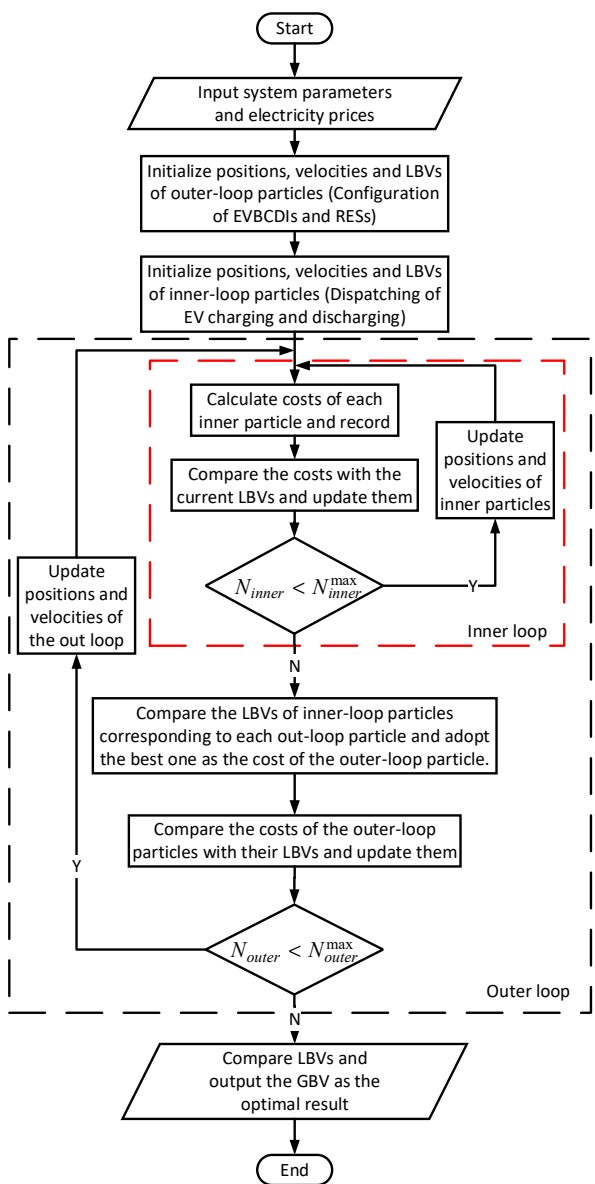

**Figure 4.** Process of the cooperative optimization. LBV: local best value, GBV: global best value.

As shown in the figure, the process consists of two loops, which correspond to optimizations of dispatching and system allocation.

The inner-loop is optimization of system dispatching. In this process, the system operation is optimized to realize the lowest total cost while the allocation of the system is fixed.

In the outer-loop, the allocation of EVs and RESs is optimized. By comparing the optimal total cost of different allocation cases, the allocation with the lowest total cost is adopted as an optimal allocation.

By employing the cooperative optimization method, economic benefits can be realized, and the system costs can be decreased.

## 6. Case Study and Discussion

### *6.1. Case Description*

#### 6.1.1. Case 1

In this case, neither numbers of EV charging/discharging infrastructures nor installation capacities of renewable energy sources are optimized. EVs will be discharged within the limits of SOCs in OBMG and be charged as much as possible when they return to RMGs. Economic dispatching of a multi-microgrid system is optimized by general management system (GMS). Case 1 is adopted as a reference in comparison to other cases.

#### 6.1.2. Case 2

In this case, the numbers of EV charging/discharging infrastructures in different microgrids are optimized while the installation capacities of RESs are fixed. Similarly, economic dispatching of multi-microgrid system is optimized by GMS.

#### 6.1.3. Case 3

In this case, the numbers of EV charging/discharging infrastructures are fixed while the installation capacities of RESs are optimized. Similarly, economic dispatching of multi-microgrid system is optimized by GMS.

#### 6.1.4. Case 4

In this case, economic dispatching and capacity allocation of RESs and EVs in multi-microgrid system are optimized synergistically. EVs and RESs can cooperate best in this case, and there is no redundant installation of EVBCDIs or RESs.

### *6.2. Simulation System Construction*

For simplification of the simulation, a multi-microgrid system with two RMGs and one OBMG is constructed. All EV owners are living in the two RMGs, identified as RMG1 and RMG2, and working in the OBMG. Exchanging power limit of the RMGs is $-95 \text{ kW} \leq P_i^G(t) \leq 95 \text{ kW}$, and the limit for OBMG is $-250 \text{ kW} \leq P_i^G(t) \leq 250 \text{ kW}$. Since the three microgrids are geographically near, PV generation and wind generation curves are similar and are forecasted by artificial neural networks using historical weather data derived from local weather records. Load demands of the three microgrids are derived from data of real microgrids with similar sizes.

A time-of-use electricity price is adopted as the energy exchanging prices with the main grid. The prices are demonstrated in Figure 5 [23]. In order to guarantee the benefits of the EV owners, charging and discharging prices of EVs should be restricted, as in (18)

$$p_C^{EV}(t) \leq \eta_C \cdot \eta_D \cdot p_D^{EV}(t) \tag{18}$$

where $p_C^{EV}(t)$ and $p_D^{EV}(t)$ are charging and discharging prices of EVs. They are assumed to be time-invariable in this paper. Additionally, to encourage participation of microgrids, discharging

prices of EVs should be lower than the highest exchanging price in OBMG, and charging prices should be higher than the lowest exchanging price in RMG.

$$\begin{cases} p_C^{EV}(t) \geq \min(p_{RMG}^G(t)) \\ p_D^{EV}(t) \leq \max(p_{OBMG}^G(t)) \end{cases} \tag{19}$$

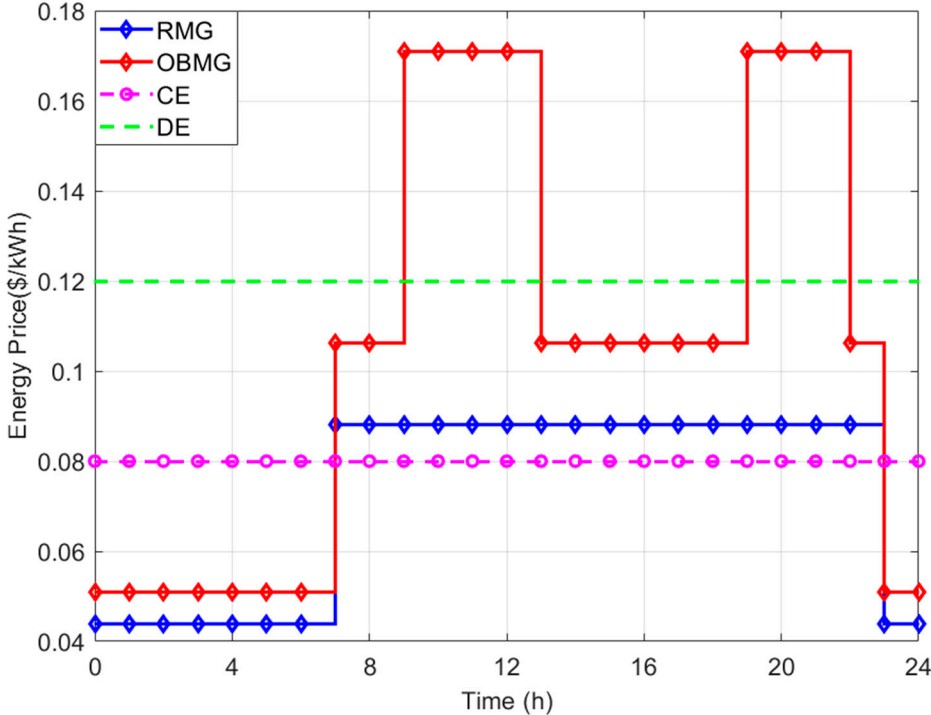

**Figure 5.** Prices of energy exchanging and EV charging and discharging.

For the EV model, BYD E6 is selected in the simulation. The capacity of the EV battery is 80 kWh and charging and discharging power limits are both 7 kW in normal charging mode. Charging and discharging efficiency are also identical, at 0.9. The daily cost of installing an EVBCDI is 0.45$. Some other data are referred to in [25].

*6.3. Simulation Results*

6.3.1. Case 1

In case one, the numbers of EVBCDIs and installation capacities of RESs are set to a fixed number according to the load demand and neither of them are optimized. The numbers of EVCDIs for RMG1, RMG2 and OBMG are 10, 10, and 20, respectively. WT installation capacities for RMG1, RMG2, and OBMG are 240 kW, 160 kW, and 320 kW. PV installation capacities for RMG1, RMG2, and OBMG are 180 kW, 120 kW, and 240 kW. The two-day operation profiles of RMG1 and OBMG in case one are shown in Figure 6. Profiles of RMG2 are similar to RMG1 and are not demonstrated here.

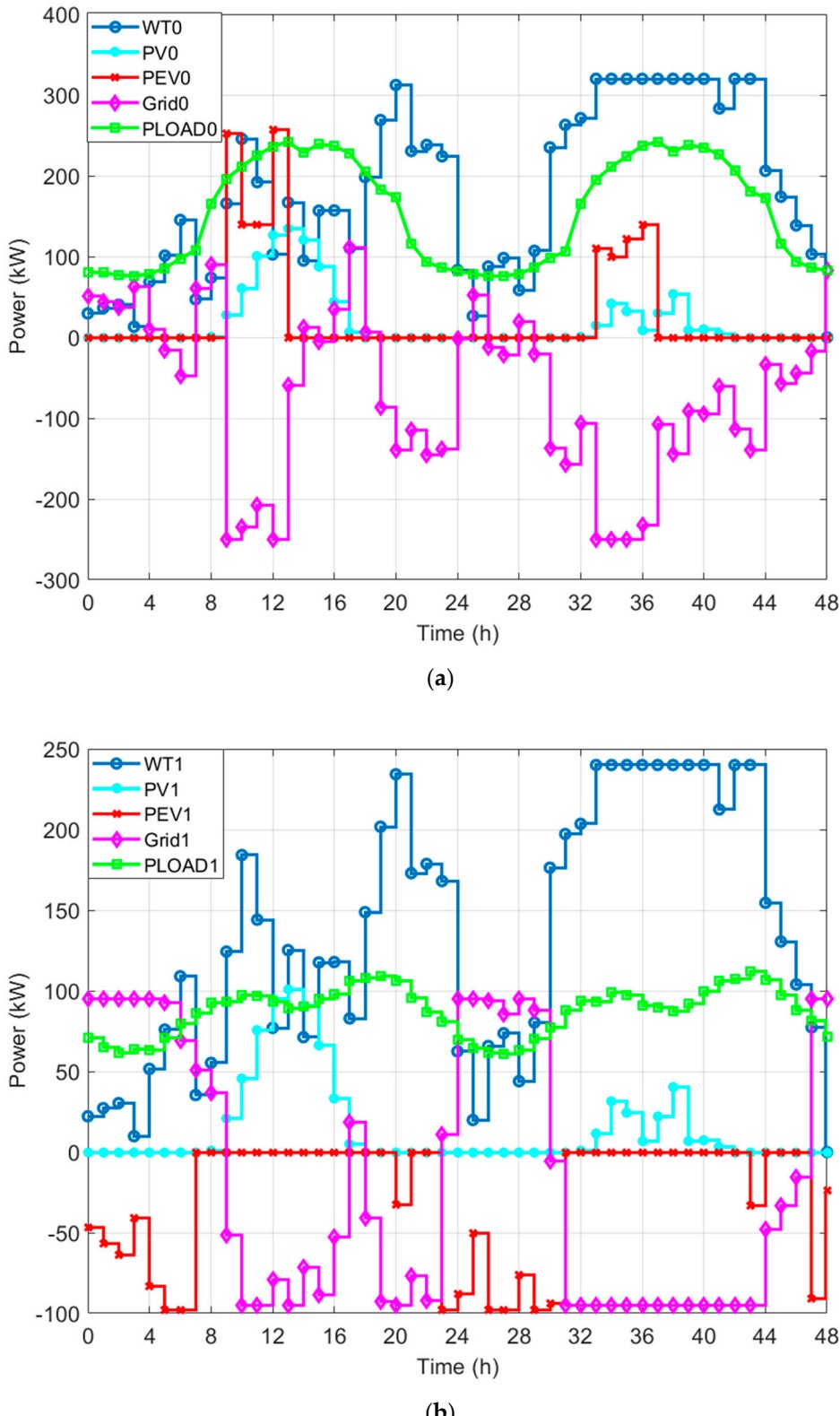

**Figure 6.** Dispatching results in Case 1: (**a**) Dispatching profiles of RMG1; (**b**) Dispatching profiles of OBMG.

As demonstrated in Figure 6, the EVs tend to charge in RMGs to satisfy their driving demands for the next day, and discharge in the OBMG to participate in the operation of the microgrid and earn some benefit by trading energy with the microgrid operator.

For most systems, the electricity prices of the main grid are always related to the load demand. The electricity prices are high in load peak hours, and the prices are low in load valley hours [26]. However, as shown in Figure 6, the microgrids tend to sell energy to the main grid when the electricity prices are high, and purchase energy from the main grid the prices of electricity are low. Therefore, the energy trading among microgrids and the grid can contribute to peak shaving of the grid.

In Case 1, the total cost of one year is $195,195.80. The costs for RMG1, RMG2, and OBMG are $67,872.80, $38,063.00, and $89,260.00, respectively.

### 6.3.2. Case 2

In Case 2, the numbers of EVBCDIs in three microgrids are optimized, while installation capacities of RESs are kept the same as Case 1. The optimal numbers of EVCDIs for RMG1, RMG2, and OBMG are 14, 15, and 29. WTs installation capacities for RMG1, RMG2, and OBMG are 240 kW, 160 kW, and 320 kW. PV installation capacities for RMG1, RMG2, and OBMG are 180 kW, 120 kW, and 240 kW. Since the two-day operation profiles of RMG1 are identical to those in Case 1, the operation profiles of RMG2 are shown here. The two-day operation profiles of RMG2 and OBMG in Case 2 are shown in Figure 7.

Tendency of EV charging and discharging is similar with Case 1. However, the numbers of EVCDIs in both RMGs and OBMG are optimized. Comparing Figure 7 with Figure 6, it is obvious that the total charging and discharging power of EVs increased. The total energy transmitted between RMGs and OBMG increased, and more energy with lower prices are transported to the places and times with higher prices. Therefore, the operation cost of the system decreased, and the total cost also decreased.

In Case 2, total cost of one year is $193,867.50. The costs for RMG1, RMG2, and OBMG are $67,491.10, $37,374.50, and $89,001.90, respectively.

### 6.3.3. Case 3

In Case 3, the numbers of EVBCDIs in three microgrids are fixed to be the same with Case 1, and installation capacities of RESs are optimized. The numbers of EVCDIs for RMG1, RMG2, and OBMG are 10, 10, and 20 respectively. Optimal WT installation capacities for RMG1, RMG2, and OBMG are 80 kW, 40 kW, and 360 kW. The installation capacity of PV modules for RMG1, RMG2, and OBMG are 180 kW, 0 kW, and 400 kW. The 2-day operation profiles of RMG1 and OBMG in Case 3 are shown in Figure 8.



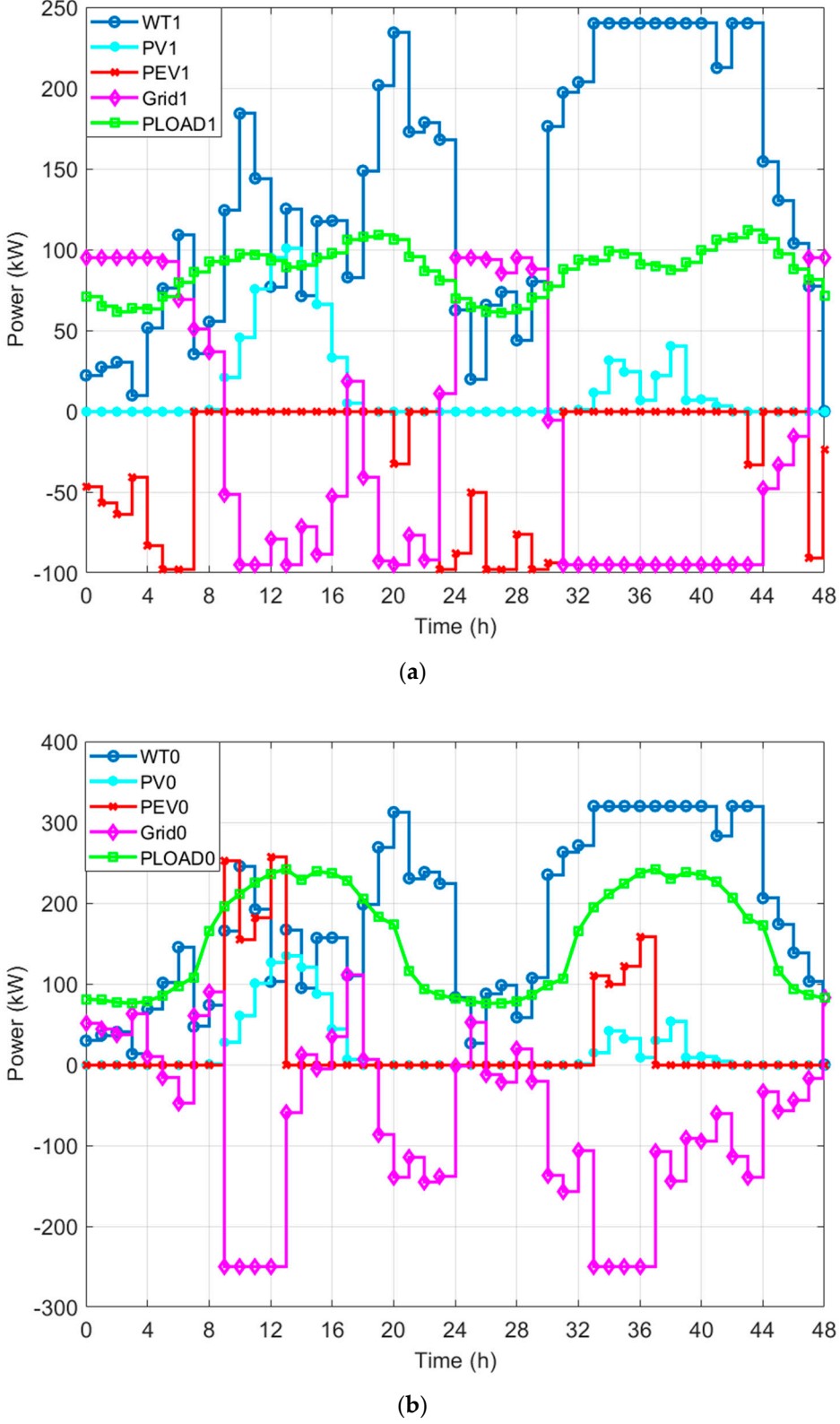

(**a**)

(**b**)

**Figure 7.** Dispatching results in Case 2: (**a**) Dispatching profiles of RMG1; (**b**) Dispatching profiles of OBMG.

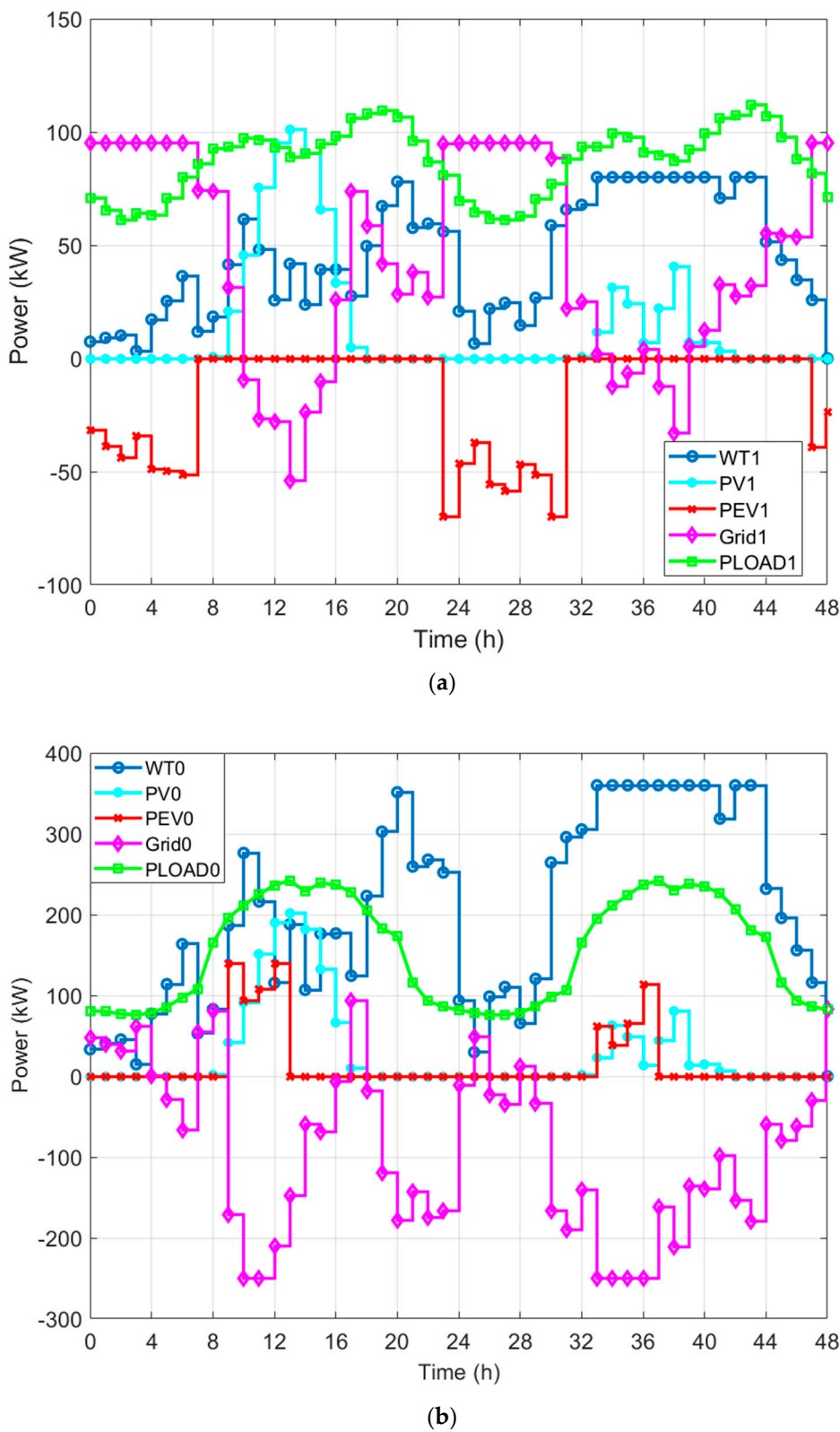

(**a**)

(**b**)

**Figure 8.** Dispatching results in Case 3: (**a**) Dispatching profiles of RMG1; (**b**) Dispatching profiles of OBMG.

According to the figures, we can see that the profiles of RMG1 are not much different. The installation capacities of RESs in RMGs and OBMG have been optimized. For the RMGs, electricity prices of the distribution grid are low and it is not beneficial to sell energy to the grid. Therefore,

the redundant installation of RESs is decreased, and less energy are sold to the grid while the load demands of RMGs are still satisfied. For OBMG, the electricity prices of the distribution grid are high, and it is beneficial to sell extra energy to the grid. Therefore, more RESs are installed to sell more energy to the grid, and the microgrid earns more from selling energy.

In Case 3, the total cost of one year is \$161,090.30. The costs for RMG1, RMG2, and OBMG are \$55,138.10, \$36,348.80, and \$69,603.40, respectively.

### 6.3.4. Case 4

In Case 4, both the numbers of EVBCDIs of the three microgrids and installation capacities of RESs are optimized. The numbers of EVCDIs for RMG1, RMG2, and OBMG are 19, 8, and 27, respectively. WTs installation capacities for RMG1, RMG2, and OBMG are 80 kW, 80 kW, and 360 kW. PV installation capacities for RMG1, RMG2, and OBMG are 120 kW, 0 kW, and 400 kW. The two-day operation profiles of RMG1 and OBMG in Case 1 are shown in Figure 9.

In this case, both EV charging devices and RESs are optimally allocated. Since the cost of PV is higher than wind generation and since the outputs of PV modules are unstable, the installation of PV modules is decreased. Meanwhile, the number of EVCDIs and the installation capacities of RESs are optimally matched. Neither redundant RES installation nor redundant installation of EVCDIs exists. Therefore, the total cost in this case is the lowest.

In Case 4, the total cost of one year is \$159,264.80. The costs for RMG1, RMG2, and OBMG are \$56,012.30, \$36,306.40, and \$66,946.10, respectively.

### 6.4. Comparison

Optimization results of system allocation are demonstrated in Table 1. Results of costs and the net exchanged energy between microgrids and the main grid in four cases are demonstrated in Table 2. The total costs and costs of the three microgrids in the four cases are listed. In addition, the costs of allocation and dispatching are also calculated and listed in the table. According to the figures in Section 6.3 and the data in the tables, conclusions can be drawn as follows:

1.  After optimization, the net exchanged energy between microgrids and the main grid decreases, which means that the microgrids becomes more independent from the main grid.
2.  Microgrids purchase energy from the main grid in load valley hours when electricity prices are low and sell energy to the main grid in load peak hours when prices are high, which contributes to the peak shaving of the main grid.
3.  In Cases 3 and 4, numbers of PVs in RMGs dramatically decrease, while number of PVs in OBMG is greatly increased. The reason is that for RMGs, the EV charging load is mainly distributed during night and the output of PVs at night is zero. Therefore, PV modules in RMGs are reduced to save installation costs.
4.  The total costs in Cases 2 and 3 are lower than that in Case 1. In Case 2, the numbers of EVBCDIs are optimized. Since redundant installation of EVBCDIs is cut down, the allocation is more proper and total cost is reduced. Similarly, in Case 3, the installation of RESs is optimized. The installation capacities of RESs are optimally configured according to the demand of the different kinds of microgrids, and the total cost is decreased compared to Case 1.
5.  In Case 4, economic dispatching and capacity allocation of RESs and EVs in the multi-microgrid system are optimized synergistically. Though the allocation cost of RESs is slightly increased, the dispatching cost of the system is significantly reduced because of the cooperative optimization of economic dispatching and capacity allocation for RESs and EVs. The total cost in Case 4 is the lowest of all four cases. The integration efficiency of RESs and EVs is improved, and more environmental benefits are achieved.

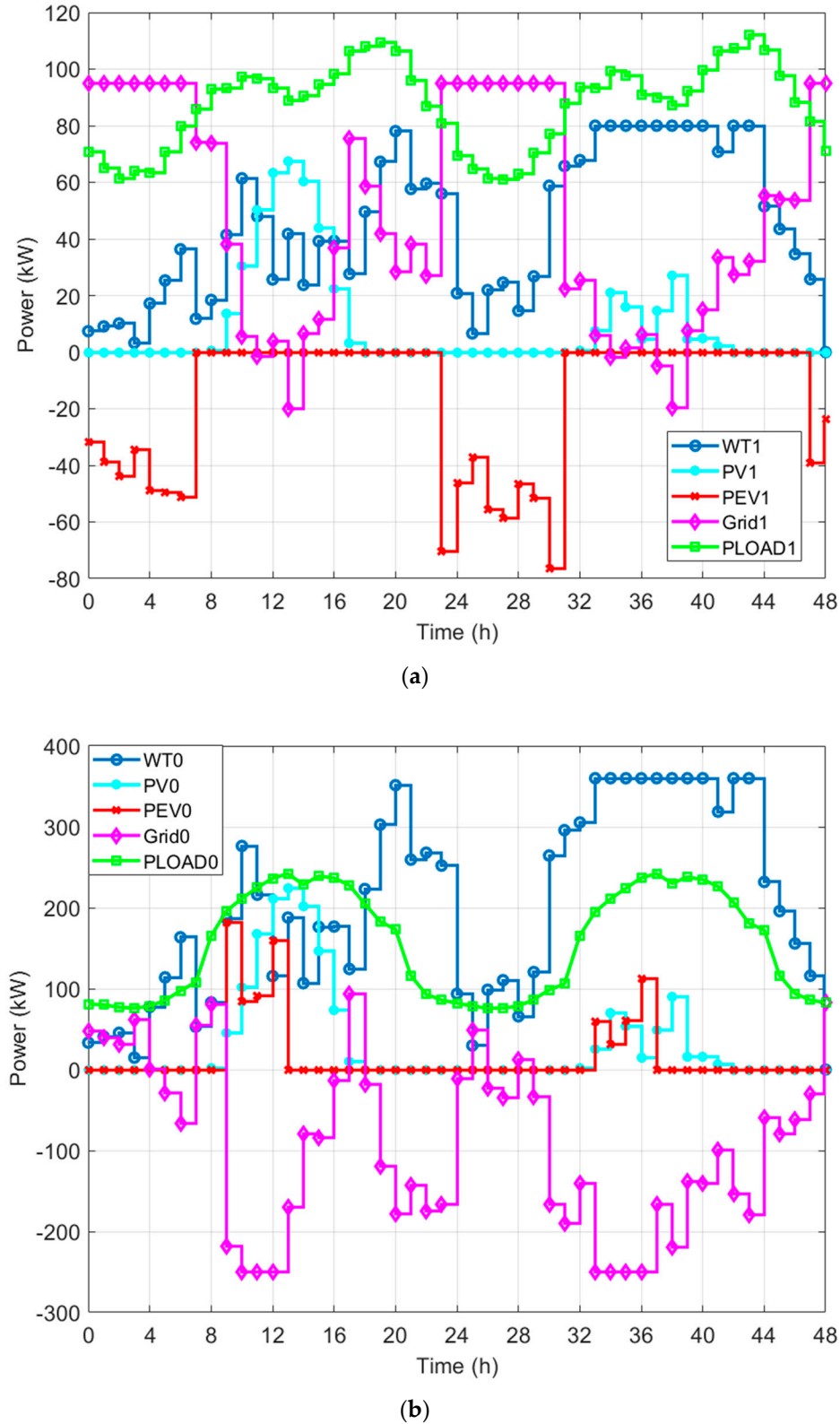

**Figure 9.** Dispatching results in Case 4: (**a**) Dispatching profiles of RMG1; (**b**) Dispatching profiles of OBMG.

**Table 1.** Comparison of Allocations in all Four Cases.

| | | Case 1 | Case 2 | Case 3 | Case 4 |
|---|---|---|---|---|---|
| EVs | RMG1 | 10 | 14 | 10 | 14 |
| | RMG2 | 10 | 15 | 10 | 12 |
| WTs | RMG1 | 6 | 6 | 2 | 2 |
| | RMG2 | 4 | 4 | 1 | 2 |
| | OBMG | 8 | 8 | 9 | 9 |
| PVs | RMG1 | 9 | 9 | 9 | 6 |
| | RMG2 | 6 | 6 | 0 | 0 |
| | OBMG | 12 | 12 | 18 | 20 |
| Allocation cost of EVs ($) | | 6570.0 | 9526.5 | 6570.0 | 8541.0 |
| Allocation cost of RESs ($) | | 88,640.9 | 88,640.9 | 70,174.1 | 72,020.8 |
| Total cost of allocation ($) | | 95,210.9 | 98,167.4 | 76,744.1 | 80,561.8 |

**Table 2.** Comparison of Costs and Net Exchanged Energy in all Four Cases.

| | Case 1 | Case 2 | Case 3 | Case 4 |
|---|---|---|---|---|
| Cost of RMG1 ($) | 67,872.8 | 67,491.1 | 55,053.9 | 56,012.3 |
| Cost of RMG2 ($) | 38,063.0 | 37,374.5 | 36,345.5 | 36,306.4 |
| Cost of OBMG ($) | 89,260.0 | 89,001.9 | 69,793.4 | 66,946.1 |
| Net Exchanged Energy (MWh) | 413.32 | 377.10 | 255.30 | 189.30 |
| Dispatching cost ($) | 97,028.4 | 95,700.1 | 84,448.7 | 78,703.0 |
| Total Cost ($) | 195,195.8 | 193,867.5 | 161,192.8 | 159,264.8 |

## 7. Conclusions

In this paper, a cooperative optimization method for a multi-microgrid system was proposed. Mathematical models of multi-microgrid systems including RESs and EVs were constructed. Energy cooperation of microgrids in the multi-microgrid system was accomplished via the mobile energy storage of EVs. The impact of ATSET of EVs on economic dispatching and capacity allocation of a multi-microgrid system was analyzed. A two-loop optimization methodology that considered the cooperation of economic dispatching and system allocation was proposed, and an IPSO algorithm was used to solve the optimization problem. Both EVs and RESs were succesfully optimized in the optimization. Four cases were presented to verify the cooperative optimization. By comparing the results of the four cases, the following conclusions are drawn:

1. By ATSET of EVs, energy can be transmitted through different microgrids, and economic dispatching of microgrids in the regional multi-microgrid system is cooperatively optimized. Optimized energy exchanging between regional multi-microgrid system and the main grid contributes to the peak shaving of the main grid.
2. By optimizing both economic dispatching and capacity allocation of RESs and EVs, the total cost of the regional multi-microgrid system is dramatically reduced, and economic benefit is achieved. After a cooperative optimization, the independency of the microgrids' operation is raised and integration efficiency of RESs and EVs is improved.

It is worth noting that the simulation system discussed in this paper was significantly simplified in order to clearly demonstrate the main methodology of the paper. Future works may focus on the application of the model to more complex systems. Real systems are always more complex, with various kinds of generation and load units, and some realistic factors such as the influence of the battery's remnant capacity on the behavior of EV owners can also be considered. A real system may also include storage batteries and micro-turbines. The way of applying the proposed model on real systems is an important topic of the future works. Moreover, since the charging and discharging prices

of EVs considered in this paper are constant, a more flexible pricing system for EVs' charging and discharging is also an issue that worth studying.

**Author Contributions:** Conceptualization, J.C. and C.C.; methodology, J.C. and C.C.; software, J.C.; validation, J.C.; formal analysis, J.C. and C.C.; investigation, J.C. and C.C.; resources, J.C. and C.C.; data curation, J.C. and C.C.; writing—original draft preparation, J.C.; writing—review and editing, J.C. and C.C.; visualization, J.C.; supervision, C.C. and S.D.; project administration, C.C. and S.D.; funding acquisition, C.C.

**Funding:** This research was funded by the National Natural Science Foundation of China, grant number 51477067 and the Lite-On Power Electronics Technology Research Fund, grant number PRC20161047.

**Conflicts of Interest:** The authors declare no conflict of interest.

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
