# Peer review of "Cooperative Optimization of Electric Vehicles and Renewable Energy Resources in a Regional Multi-Microgrid System"

_applsci, doi:10.3390/app9112267_

Round 1

Reviewer 1 Report

The authors have studied an interesting and timely problem. Considering EVs simultaneously with renewable energy sources can be used as a remedy for the existing issues in power systems. However, the reviewer has the following minor and major concerns. 

1- Uncertainty is an inseparable nature of renewable energies, why the authors have neglected the stochasticity?    

2- The introduction does not provide strong support for the contributions, more recently published works should be reviewed. 

3- Please provide some references in the introduction to show the existing gap and to highlight the contributions of your work. 

The authors may use the following references to address the first three comments. 

[A]. "Uncertainty-Based Models for Optimal Management of Energy Hubs Considering Demand Response", doi: https://doi.org/10.3390/en12081413

[B]. "Stochastic network-constrained co-optimization of energy and reserve products in renewable energy integrated power and gas networks with energy storage system", doi: https://doi.org/10.1016/j.jclepro.2019.03.021

[C]. "Optimal Selection of Navigation Modes of HEVs Considering CO2 Emissions Reduction," doi: https://doi.org/10.1109/TVT.2019.2894383

[D]. "A Stochastic Bilevel Model for the Energy Hub Manager Problem," doi: https://doi.org/10.1109/TSG.2016.2618845

[E]. "Optimal scheduling of plug-in electric vehicles and renewable micro-grid in energy and reserve markets considering demand response program", doi: https://doi.org/10.1016/j.jclepro.2018.03.058

4- Please provide proper references to the presented equations. 

5- The result section should be carefully revised. Please provide adequate explanation instead of just putting the informative figures and tables. 

6- In the conclusion section, please put the prospect of future works. 

7- The English language needs to be polished. 

Author Response

The authors are grateful of the careful review and valuable comments by the AE and reviewers on our work. This manuscript has been reviewed carefully based on your suggestions. All the respected reviewers’ concerns have been recognized in the revised manuscript and responded to in this supporting document. We look forward to your positive response.

The modifications to the manuscript are made as follows, and written in red.

(1) The abstract and the introduction have been revised to strengthen the literature survey and clarify the scope and the novelty of this manuscript.

(2) Some descriptions of the model have been improved to make it more clear, rigid and understandable.

(3) Results in case studies have been elaborated to make it more understandable.

(4) Grammar errors and typos have been corrected.

Best Regards!

Sincerely yours

Reviewer 2 Report

Author comments

The paper main scope is to present a novel cooperative optimization method for regional multi-microgrid system, optimizing dispatching strategy and capacity allocation of electric vehicles and renewable energy sources and it implies that across-time-and-space energy transmission of electric vehicles are both considered in the optimization model.

I have performed a detailed review of state of the art and form my point of view the field is extensively analyzed in the literature but the approach, considering the aspect of “cooperative optimization” is novel.

I suggest some key points to improve the global quality of the research presented:

·       3.4. Include a brief diagram to clearly explain the operation mechanism of the system, it is one of the key points to clearly understand the whole paper (similar to Figure 3).

·       Why two loops is the chosen method? It is explained but it should be justified in order to clearly analyze possible improvements of the methodology, considering additional possible optimizations.

·       Figures 4, 5, 6, 7 and 8 are difficult to read and understand, and they are of great interest to all readers. I would suggest the authors improve it because it will improve the global quality of the whole paper.

The reviewer.

Author Response

(The authors gave the same response as above.)

Round 2

Reviewer 1 Report

The authors have addressed all the existing concerns.

Author Response

The authors appreciate your time spent for reviewing this manuscript. We are grateful for this opportunity to disseminate our findings in the high-ranking journal.
